# Low-Frequency Vibration Sensor with a Sub-nm Sensitivity Using a Bidomain Lithium Niobate Crystal

**DOI:** 10.3390/s19030614

**Published:** 2019-02-01

**Authors:** Ilya V. Kubasov, Aleksandr M. Kislyuk, Andrei V. Turutin, Alexander S. Bykov, Dmitry A. Kiselev, Aleksandr A. Temirov, Roman N. Zhukov, Nikolai A. Sobolev, Mikhail D. Malinkovich, Yuriy N. Parkhomenko

**Affiliations:** 1Department of the Materials Science of Semiconductors and Dielectrics, National University of Science and Technology MISiS, 119049 Moscow, Russia; akislyuk94@gmail.com (A.M.K.); aturutin92@gmail.com (A.V.T.); xalexx1349@mail.ru (A.S.B.); dm.kiselev@gmail.com (D.A.K.); temirov.alex@yandex.ru (A.A.T.); rom_zhuk@mail.ru (R.N.Z.); sobolev@ua.pt (N.A.S.); malinkovich@yandex.ru (M.D.M.); parkh@rambler.ru (Y.N.P.); 2Department of Physics and I3N, University of Aveiro, 3810-193 Aveiro, Portugal

**Keywords:** lithium niobate, bidomain crystal, vibration, sensor

## Abstract

We present a low-frequency sensor for the detection of vibrations, with a sub-nm amplitude, based on a cantilever made of a single-crystalline lithium niobate (LiNbO_3_) plate, with a bidomain ferroelectric structure. The sensitivity of the sensor-to-sinusoidal vibrational excitations was measured in terms of displacement as well as of acceleration amplitude. We show a linear behavior of the response, with the vibrational displacement amplitude in the entire studied frequency range up to 150 Hz. The sensitivity of the developed sensor varies from minimum values of 20 μV/nm and 7 V/*g* (where *g* = 9.81 m/s^2^ is the gravitational acceleration), at a frequency of 23 Hz, to peak values of 92.5 mV/nm and 2443 V/*g*, at the mechanical resonance of the cantilever at 97.25 Hz. The smallest detectable vibration depended on the excitation frequency and varied from 100 nm, at 7 Hz, to 0.1 nm, at frequencies above 38 Hz. Sensors using bidomain lithium niobate single crystals, as sensitive elements, are promising for the detection of ultra-weak low-frequency vibrations in a wide temperature range and in harsh environments.

## 1. Introduction

Precise measurements of weak vibrations are of interest for structural health monitoring [1,2,3], security [4], and active vibration damping [5,6,7]. In order to detect a vibration, it is necessary to convert the energy of an oscillatory movement to an electrical signal. Optical and piezoelectric sensors are the most frequently used devices for this purpose [8]. The optical vibration sensors are usually more precise and sensitive but have large sizes and a high power consumption; hence they are expensive and inconvenient to be used as parts of, for example, a distributed network in complex constructions. On the contrary, devices that use the piezoelectric effect to detect vibrations are simple, small, and able to operate both in active (as surface acoustic wave structures [9]) and passive (bending, shear, or torsion mechano-electric converters [10,11,12]) regimes. Moreover, a complex vibrational sensing network can be initialized from a standby mode by a single passive piezoelectric detector [13]. It is also an advantage that the piezoelectric sensors can be designed so as to harvest mechanical energy of vibrations and transform it to electricity, thus enabling the creation of self-powered devices [14].

In spite of the fact that a large number of different shapes and designs of the piezoelectric sensitive elements were offered, only a few materials were tested. Commonly, the sensors contain lead zirconate titanate (PZT) ceramics, having different compositions and states (e.g., films [11,12,15], bulk ceramics [13,16,17,18,19,20], or fiber composites [21]), although for high-temperature applications single crystals of quartz [22], langasite [23], lithium niobate (LN) [24], yttrium oxyborate [25], and aluminum nitride films [10] were also suggested. Good reviews on this topic can be found in [8,24,26]. Despite the fact that single-crystalline piezoelectric materials possess a higher thermal stability than PZT, their main disadvantage—low values of piezoelectric coefficients—is the reason why PZT is still used in the vast majority of cases.

The problem of the weak conversion of a mechanical deformation into an electrical signal by single-crystalline piezoelectric materials can be solved by utilizing complex constructions, such as unimorphs, bimorphs, or multilayer composites [27,28,29,30,31], but the presence of adhesive layers or grain boundaries in these composite transducers decreases the sensitivity, as well as the accuracy and thermal stability of the sensors. However, there is a way to manufacture a series bimorph for the piezoelectric sensing element and avoid bonding of separate plates by the formation of two domains with oppositely-directed spontaneous polarization vectors in a ferroelectric single-crystalline plate. If the crystallographic cut is correctly selected, such a “bidomain” crystal demonstrates a bimorph-like behavior, but does not comprise any interface except for an interdomain wall. Bending deformation of this single-crystalline bimorph causes the expansion of one domain and contraction of its counterpart. The voltages induced in the domains by the direct piezoelectric effect are added up; they are proportional to the bending magnitude at a fixed frequency. An example of the ferroelectric material which can be produced in a bidomain state is lithium niobate (LiNbO_3_).

The phenomenon of the domain inversion in LiNbO_3_ in the course of a heat treatment near the Curie point was discovered by Ohnishi [32]. For the first time, large-area bidomain lithium niobate (b-LN) wafers were produced by Nakamura et al. [33,34] and then investigated by numerous researchers [33,34,35,36,37,38,39,40,41,42,43,44,45,46]. Initially, b-LN single crystals were proposed as a possible replacement of composite bimorphs, glued by epoxy resins in precise movement systems, and for energy harvesting applications [33,34,39,40,41,47,48]. Recently, we have shown the possibility to apply b-LN for the conversion of oscillatory deformations to electrical signals [49,50,51,52] in vibrational sensors as well as in laminate magnetoelectric composites (b-LN/Metglas^®^), with a record value of sensitivity to magnetic field as low as 200 fT at a frequency of 6862 Hz [51]. Lithium niobate exhibits high thermal stability of piezoelectric coefficients, elastic properties, and electromechanical coupling factors for different cuts [53,54,55], and, moreover, possesses lead-free composition, which make it a promising candidate for the substitution of the commonly used PZT ceramics in low-frequency sensing and actuation applications.

## 2. Materials and Methods

In this paper we report the use of a single-crystalline b-LN plate for detecting low-frequency mechanical vibrations. A long and narrow rectangular sample with dimensions of 75 × 5 × 0.5 mm^3^ was cut from a commercial single-domain LiNbO_3_ wafer (ELAN Company Ltd., Saint-Petersburg, Russia) with the *y* + 128° crystallographic orientation. Two oppositely polarized ferroelectric domains (so-called “head-to-head” bidomain structure) were formed in the plate by the diffusion annealing technique [32,34,42]. The detailed description of the b-LN preparation procedure can be found in Appendix A. Then tantalum electrodes were deposited on the opposite faces of the b-LN crystal by DC (direct current) magnetron sputtering. The quality of the bidomain structure was tested using a cantilever-type fastening by measuring the free end deflection under an applied external voltage, by a technique described in detail elsewhere [51]; the *k*-factor revealing the quality of the produced bidomain structure was determined to be as high as 33.1 pm/V, which is only 18% lower than the theoretically predicted value (*k*_y+128°_ = 1.5·*d*_23_ = 40.4 pm/V).

We also prepared another LN sample with the same size, crystallographic orientation, and electrode type, but having a single-domain ferroelectric structure. This specimen was used for the evaluation of the external electromagnetic noise floor.

Finally, the b-LN crystal and its single-domain counterpart were clamped by two stainless-steel screws with nuts as a cantilever, with a length of 70 mm in a home-made fastening tool containing a polycrystalline alumina base, two gaskets, and two clamps. Two strips made of aluminum foil pressed by clamps to the tantalum electrodes were used for transferring the generated voltage to a coaxial cable and then to the measuring system (Figure 1).

In order to excite vibrations with ultra-low magnitude and frequency (down to 0.1 nm and 1 Hz, respectively) we used a home-made piezoelectric shaker based on two similar PZT tubes (ceramic type APC 850, APC International Ltd., Mackeyville, PA, USA), with a length of 40 mm, and inner and outer diameters of 10 and 11 mm, respectively. These tubes were placed vertically on a massive steel plate fixed on a pneumatically stabilized optical table (Standa Ltd., Vilnius, Lithuania). The sensor prototype was mounted on a light aluminum platform fastened on the top of the PZT tubes. Finally, the sensor was shielded by a grounded copper box to reduce the electromagnetic noise from external sources.

Mechanical vibrations were excited by applying an AC voltage from an external signal generator to the PZT tubes connected in parallel. In the present study we used only pure sine excitations with low magnitudes and frequencies (less than 11.5 V and 150 Hz, respectively), so that the excitation of the PZT tubes was always linear with respect to the applied ac signal.

The measurements of sensitivity to the vibrational impact were carried out at room temperature in an air ambient. We used two different techniques for the measurement of the sensor response to the external mechanical vibrations: (i)Lock-in detection of the voltage amplitude by a SR-830 amplifier (Stanford Research Systems Inc., Sunnyvale, CA, USA);(ii)Registration of oscillograms by a DSO-X 3032A oscilloscope (Agilent Technologies Inc., Santa Clara, CA, USA) with post-processing by the Fourier analysis.

Both the techniques were implemented without any additional preamplification of the initial signal collected from the b-LN sensitive element. As a consequence, the impedances of the sensor and measuring instrument were not matched. For this reason, the amplitude values of the signals collected by the setups were corrected with respect to the impedances of the measuring instrument, cables, and samples in order to compute the open-circuit voltage generated by the sensitive b-LN element.

The description of the used setup configurations, impedance measurements, as well as the details of the correction of the collected signals are presented in Appendix B.

The measurements of the sensor sensitivity by method (i) were carried out at four different excitation amplitudes. The voltages corresponding to vertical movements of the sensor amounting to 0.1, 1, 10, and 100 nm were applied to the PZT tubes of the shaker (note that here we operate with amplitude values, so that the peak-to-peak oscillations were twice as large). Method (ii) was implemented by applying a sine voltage signal with an amplitude of 11.5 V to the PZT tubes, which corresponds to a 161 nm oscillation. In addition, the electromagnetic and acoustic noise floors were measured on the single-domain and bidomain crystals, respectively.

## 3. Results and Discussion

The results of the sensitivity measurements are shown in Figure 2. The generated voltage was relatively strong even at the minimal displacement amplitude of 0.1 nm, and exceeded the acoustic noise level in almost the entire frequency range studied. Indeed, sine vibrations with displacement amplitudes of 0.1, 1, 10, and 100 nm could be confidently detected at frequencies higher than 38, 23, 14, and 7 Hz, respectively. The acoustic noise floor measured from the sensor without any excitation was lower than 10 μV in almost the entire frequency range. There were two sharp noise peaks in the graph related to zero excitation: one was near the resonance frequency of 97.25 Hz and the other at 50 Hz, which is the power line frequency in Russia. Below approximately 30 Hz, the acoustic noise started to steadily increase with decreasing frequency, reaching a maximum value of ca. 100 μV near 7 Hz. We ascribe the signal’s behavior to some low-frequency sound source or residual mechanical vibrations. The acoustic nature of this noise source was confirmed by the electromagnetic noise floor acquired from the single-domain cantilever that demonstrated a low-frequency noise behavior (1/*f* flicker) having a different shape.

The signal measured by the oscilloscope possessed a slightly different shape in comparison to the lock-in results, especially at frequencies below 50 Hz. It was clear that a sharper line was a result of calculation errors, which are imminent when the FFT computing is utilized for finite signals in the time domain. However, the low-frequency part of the plot was mainly influenced by the low input impedance of the oscilloscope that forms a high-pass filter with the impedance of the b-LN crystal.

A mechanical quality factor calculated for the resonance peak was ca. 328 for measurements with all intensities of the vibrational excitation used in the present work.

The dependencies of the generated voltage on the displacement amplitude were linear in almost the entire investigated range (Figure 3a). An exception from this trend was only the data obtained in the oscilloscope measurements; the main reason for such a behavior was discussed above. As the experimental data at a fixed frequency were straight lines that differed only in their slopes, we could get the sensitivity to the vibrational amplitude normalized to a 1 nm displacement simply by calculating a linear fit for each frequency and taking the first-order term (Figure 3b). The plot of the sensitivity to a 1 nm displacement confidently followed the graph of the signal, which was obtained at a 1 nm excitation (Figure 2) down to a frequency of 23 Hz, where the sensitivity reached its minimum value of 20 μV/nm. A discrepancy at low frequencies was associated with an increasing contribution of the noise and, as a consequence, a worse linear fit. The peak value of the sensitivity to oscillatory displacements was reached at the resonance frequency and equaled 92.5 mV/nm.

Finally, we computed the sensitivity of our sensor to the acceleration amplitude in units of the gravitational acceleration *g* = 9.81 m/s^2^. Due to the weak intensity and low frequency of the used driving signals and, consequently, the high accuracy of the transformation of the AC (alternating current) voltage into oscillating deflections, the vibration at a fixed frequency had a strongly sinusoidal character. This means that the acceleration sensitivity could be easily calculated just by dividing the displacement sensitivity by a factor of 4·*π*^2^·*f*^2^·*g*^−1^, as the acceleration is the second derivative of the displacement. Figure 4 shows that the highest sensitivity of the sensor reached a value of 2443 V/*g* at a resonance frequency of 97.25 Hz, and decreased to ca. 7 V/*g* to the left and to the right from the maximum point.

## 4. Conclusions

To conclude, a vibrational sensor based on a bidomain LiNbO_3_ (b-LN) single crystal was developed and investigated. The sensitive element was made of a rectangular b-LN plate fastened as a cantilever in a polycrystalline alumina clamp. We used a home-made piezoelectric shaker, as well as two different voltage measurement setups based on a lock-in and on an oscilloscope, respectively. The sensor demonstrates the ability to detect sine vibrations with displacement amplitudes down to 0.1 nm at frequencies higher than 38 Hz without any preamplification. Vibrations with a larger amplitude of, for example, 100 nm allow the detection limit to be pulled down to a frequency of 7 Hz. Moreover, the sensor’s output voltage at a fixed frequency demonstrates a linear response to the increasing displacement amplitude. The displacement response to low-amplitude sine vibrational excitations was measured to be equal to 92.5 mV/nm at the resonance frequency of 97.25 Hz, and to 20 μV/nm at 23 Hz. The obtained maximum value of the acceleration sensitivity reaches 2443 V/*g* when the sensor is oscillatory excited at resonance. Our device is quite competitive with the sensors based on PZT, polyvinylidene difluoride (PVDF), or ZnO [11,14,18,19,20,21,56]. It possesses the same or a higher sensitivity as compared to the best results published (2 mV/nm at a resonance frequency of 48 Hz for a cantilever based on a PZT fiber composite [21], and 170 V/*g* at a resonance frequency of 53.6 Hz for a cantilever based on a PZT bulk ceramic [19]). A more detailed comparison with the literature data can be found in Appendix C.

The high thermal and chemical stability of lithium niobate, as well as an efficient conversion of mechanical deformations to voltage, makes the b-LN crystals a promising material for highly sensitive applications, including low-frequency vibrational sensors able to withstand harsh environments and high temperatures.

## Figures and Tables

**Figure 1 sensors-19-00614-f001:**
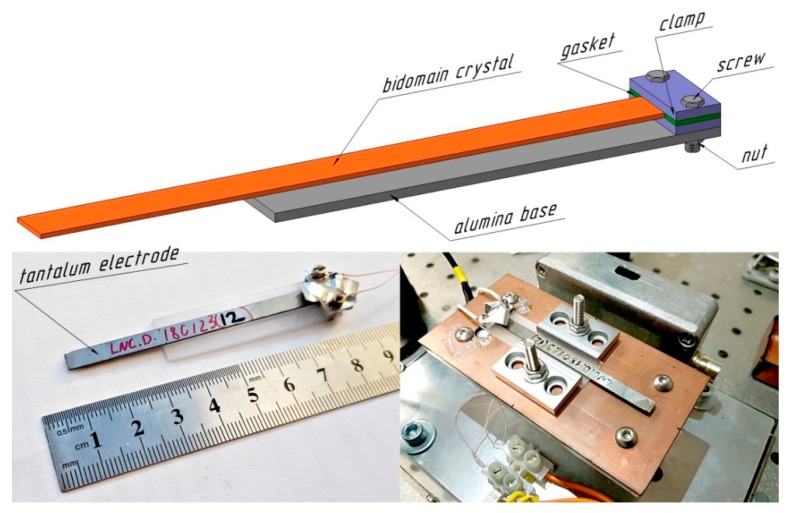
Schematic representation (**top**) and photography of the sensor prototype alone (**bottom left**) and mounted on the shaker (**bottom right**); the shielding box was removed. The labeling on the bidomain crystal was added for its identification.

**Figure 2 sensors-19-00614-f002:**
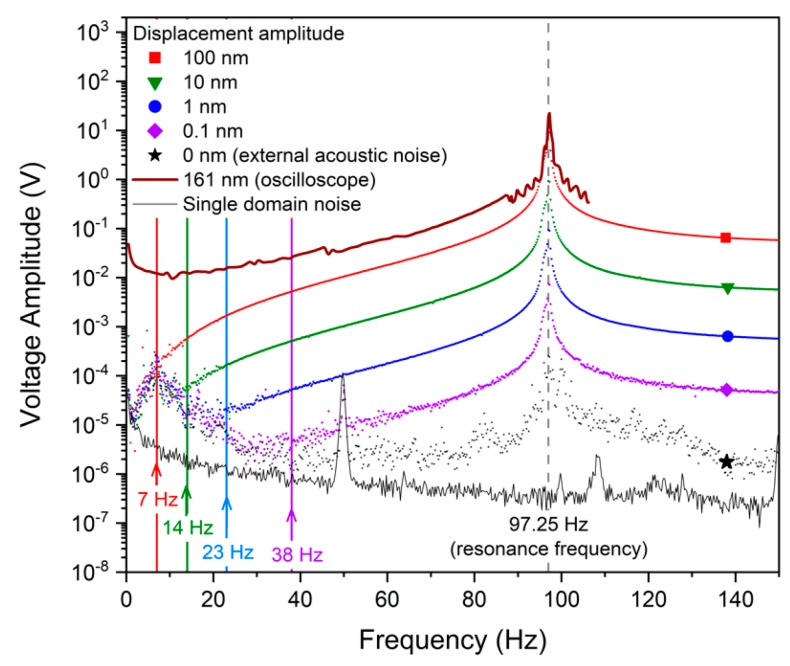
Voltage generated by the sensor being subject to sine vibrational excitations with different displacement amplitudes.

**Figure 3 sensors-19-00614-f003:**
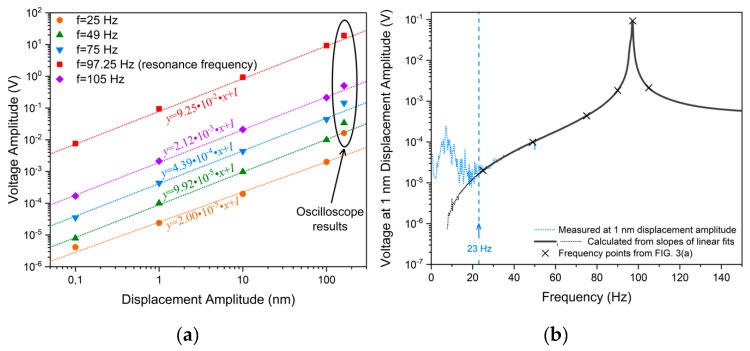
(**a**) Voltage generated by the sensor upon sine vibrations as a function of the displacement amplitude (the *I* terms in the line equations in panel (**a**) are points of intercept defined by the acoustic noise; *I* < 5 μV for all linear graphs shown); (**b**) sensitivity plot representing slopes of the linear responses to vibrations at all investigated frequencies (oscilloscope results neglected) and compared with the data for a 1 nm excitation displacement amplitude.

**Figure 4 sensors-19-00614-f004:**
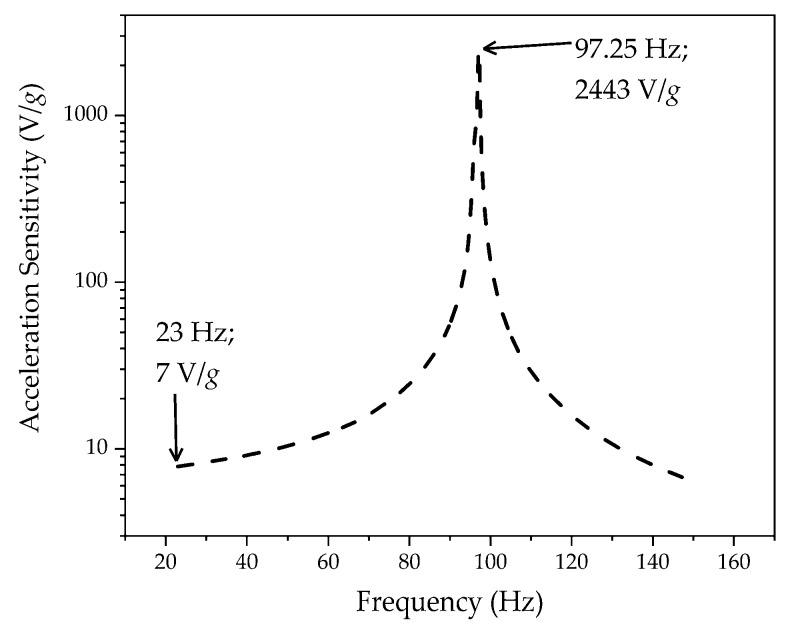
Sensitivity of the sensor to acceleration in units of *g*.

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
