# Peer review of "Low-Frequency Vibration Sensor with a Sub-nm Sensitivity Using a Bidomain Lithium Niobate Crystal"

_sensors, 2019, doi:10.3390/s19030614_

Round 1

Reviewer 1 Report

The authors propose a bidomain lithium niobate crystal as a low frequency vibration sensor with a sub-nanometer sensitivity. The proposed device is less power-consuming than its optical counterparts, and it shows the advantages of bimorph configurations while avoiding the bonding of separate plates. This is achieved by providing a bidomain structure, made of two opposite ferroelectric domains. The approach is innovative and shows a sensitivity as low as 0.1 nm at frequencies higher than 38 Hz. Such a sensor can be of high interest for health monitoring or vibration damping.

However, we recommend minor revision before considering publication:

-          At the end of the introduction, the authors say b-LN show high thermal stability. Actually, it is true that LN can withstand very high temperature (> 1000°C), the Curie temperature being itself very high (1133°C). But the k factor can significantly change with temperature, which can deteriorate the stability of the sensitivity. As temperature stability is mentioned as an argument in favor of lithium niobate, I would suggest complementary measurements showing the behavior of k as a function of temperature. Otherwise, I would suggest to change the end of the introduction.

-          The sensor is made of two oppositely polarized ferroelectric structures made through diffusion annealing techniques. The description of the manufacturing process is unclear. More information is mandatory:  what is the material diffused inside the crystal?  What is the diffusion depth? What are the temperature diffusion and the duration of diffusion?

Such information is necessary to understand the performances of the device.

Author Response

First of all, the authors would like to thank Reviewer 1 for his/her thoughtful suggestions, comments and criticisms. We tried to address all of them in the revised manuscript and we believe that it has significantly improved.

Point 1: At the end of the introduction, the authors say b-LN show high thermal stability. Actually, it is true that LN can withstand very high temperature (> 1000°C), the Curie temperature being itself very high (1133°C). But the k factor can significantly change with temperature, which can deteriorate the stability of the sensitivity. As temperature stability is mentioned as an argument in favor of lithium niobate, I would suggest complementary measurements showing the behavior of k as a function of temperature. Otherwise, I would suggest to change the end of the introduction. 

Response 1: Lithium niobate is an oxide material with high melting point (1257°C) and Curie temperature, so that it can withstand very high temperatures in an air ambient without any change of the ferroelectric domain structure. Moreover, the piezoelectric properties, coefficients, elastic properties and electromechanical coupling factors for different crystal cuts almost do not change at least up to 500°C (see, e.g., DOI: 10.1016/J.JMAT.2018.10.001). Consequently, the static k-factor that is mentioned in the manuscript is almost constant too, as the piezoelectric coefficient d23 remains constant or even slightly grows (DOI: 10.1143/JJAP.6.151). In fact, the thermal stability of the discussed b-LN sensing element is determined by the electrical conductivity of the material, which starts to decrease dramatically above 450...500°C. It is a much wider temperature range than a PZT sensor can offer. At the end of the introduction, we additionally added references to papers which confirm the thermal stability of lithium niobate.

Point 2: The sensor is made of two oppositely polarized ferroelectric structures made through diffusion annealing techniques. The description of the manufacturing process is unclear. More information is mandatory: what is the material diffused inside the crystal? What is the diffusion depth? What are the temperature diffusion and the duration of diffusion?

Response 2: More detailed information about the b-LN preparation procedure by diffusion annealing is added in Appendix A.

Reviewer 2 Report

The authors present a vibration sensor composed of a LiNbO3 cantilever made of two ferroelectric domains with opposite spontaneous polarizations. The sensor is designed to obtain high sensitivity in the low frequency range (few Hz to 150Hz). It is presented as a viable alternative to PZT made sensors but with greater thermal stability and with the possibility to be used at high temperature.

Detailed characterizations of the fabricated device are presented. Robust data are obtained using two measurement methods (direct measurements and FFT). The developed sensor has displacement sensitivity close to 100 mV/nm and acceleration sensitivity of 2500V/g at the resonance. In addition, measurable displacement amplitude as low as 0.1 nm can be detected. These performances are competitive with published results based on other architectures and platforms.

Although extensive references are given it would help the reader to add few sentences to explain both the fabrication of the bi-domain crystal and the physical principle of the bi-domain LiNbO3 sensor. In addition, characteristics of the structure of the fabricated bi-domain are missing. Can the sensor response be further improved by changing the geometry or the fabrication process ?

In the appendix it seems that equation A3 has to be corrected : cW2 should appear instead of cw1 ?

The paper technical content is well written and interesting for the community. If the above-mentioned remarks/questions are clarified, the paper can be published in Sensors journal.

Author Response

First of all, the authors would like to thank Reviewer 2 for his/her thoughtful suggestions, comments and criticisms. We tried to address all of them in the revised manuscript and we believe that it has significantly improved.

Point 1: Although extensive references are given it would help the reader to add few sentences to explain both the fabrication of the bi-domain crystal and the physical principle of the bi-domain LiNbO3 sensor. In addition, characteristics of the structure of the fabricated bi-domain are missing. Can the sensor response be further improved by changing the geometry or the fabrication process ? 

Response 1: More detailed information about the b-LN preparation procedure by diffusion annealing is added in Appendix A. The explanation of the working principle of the bimorph b-LN sensing element was added in the Introduction. However, ways of improving the sensor’s response by changing the geometry or the fabrication process were not investigated in the present study. We just can suggest choosing a more appropriate geometry for a specific application, e.g., by increasing the length and decreasing the thickness of the sensing element one can decrease the bending resonance frequency.

Point 2: In the appendix it seems that equation A3 has to be corrected: cW2 should appear instead of cw1?

Response 2: Thank you for the correction! We fixed the misprint in equation A3.

Reviewer 3 Report

As to sub nanometer sensor, you should give more data about sub nanometer instead of 100/10nm. 

The sensor can measure weak vibation with high precision. In fact, I dont find any great innovation to improve its precision. And the work need further investigateion.

Author Response

First of all, the authors would like to thank Reviewer 3 for his/her thoughtful suggestions, comments and criticisms. We tried to address all of them in the revised manuscript and we believe that it has significantly improved.

Point 1: As to sub nanometer sensor, you should give more data about sub nanometer instead of 100/10nm.

Response 1: The data which proves the sensitivity to sub-nm vibrations is shown at Figure 2 and Figure 3 (a) of the manuscript. In spite of the relatively high detection frequency of such vibrations (38 Hz and above) we can claim that the sensor does demonstrate the ability to detect them.

Point 2: The sensor can measure weak vibration with high precision. In fact, I dont find any great innovation to improve its precision. And the work need further investigation.

Response 2: The comment does not contain any specific claim to the manuscript. Further investigations leading to improved sensor characteristics are indeed under way and will be published in forthcoming publications.
